# Compact Language Models via Pruning and Knowledge Distillation

**Saurav Muralidharan**[*]   **Sharath Turuvekere Sreenivas**[*]   **Raviraj Joshi**
**Marcin Chochowski**   **Mostofa Patwary**   **Mohammad Shoeybi**   **Bryan Catanzaro**
**Jan Kautz**   **Pavlo Molchanov**
NVIDIA
{sauravm,sharatht,ravirajj,mchochowski,mpatwary,mshoeybi,
bcatanzaro,jkautz,pmolchanov}@nvidia.com

## Abstract

Large language models (LLMs) targeting different deployment scales and sizes are currently produced by training each variant from scratch; this is extremely compute-intensive. In this paper, we investigate if pruning an existing LLM and then re-training it with a fraction (<3%) of the original training data can be a suitable alternative to repeated, full retraining. To this end, we develop a set of practical and effective **compression best practices** for LLMs that combine depth, width, attention and MLP pruning with knowledge distillation-based retraining; we arrive at these best practices through a detailed empirical exploration of pruning strategies for each axis, methods to combine axes, distillation strategies, and search techniques for arriving at optimal compressed architectures. We use this guide to compress the Nemotron-4 family of LLMs by a factor of 2-4×, and compare their performance to similarly-sized models on a variety of language modeling tasks. Deriving 8B and 4B models from an already pretrained 15B model using our approach requires up to 40× fewer training tokens per model compared to training from scratch; this results in compute cost savings of 1.8× for training the full model family (15B, 8B, and 4B). MINITRON models exhibit up to a 16% improvement in MMLU scores compared to training from scratch, perform comparably to other community models such as Mistral 7B, Gemma 7B and Llama-3 8B, and outperform state-of-the-art compression techniques from the literature. We have open-sourced MINITRON model weights on Huggingface [2], with corresponding supplementary material including example code available on GitHub [3].

## 1 Introduction

Large language models (LLMs) now dominate real-world natural language processing and have demonstrated excellent proficiency in understanding difficult contexts [7, 40, 52, 49, 48]. To aid users targeting different deployment sizes and scales, model providers often train an entire family of models from scratch, each with a different size (number of parameters). For instance, the LLaMa-2 model family [49] includes three different variants with 7, 13, and 70 bil-

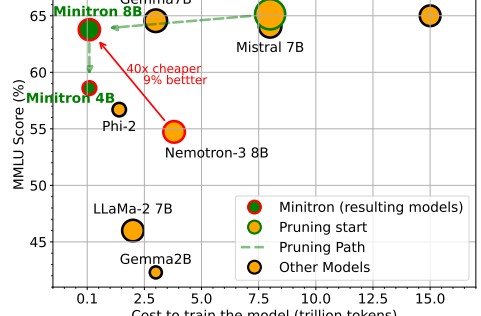

Figure 1: MINITRON results. Our approach greatly reduces training costs for *additional models* (40×) while improving accuracy.

---

[*]Equal contribution.

[2]Minitron Collection on HuggingFace
[3]https://github.com/NVlabs/Minitron

38th Conference on Neural Information Processing Systems (NeurIPS 2024).

| DEP | MLP | ATT | EMB | Distillation Loss | LM Val Loss |
|-----|-----|-----|-----|-------------------|-------------|
| ✓ | ✓ | ✓ | ✓ | 5.35 → 0.38 | 2.062 |
| ✗ | ✓ | ✓ | ✓ | **6.33 → 0.37** | **2.049** |
| ✗ | ✓ | ✓ | ✗ | 5.07 → 0.42 | 2.101 |
| ✓ | ✗ | ✗ | ✗ | 8.35 → 0.49 | 2.155 |
| Train from scratch (random init) | | | | 12.27 → 2.34 | 3.953 |

Table 1: Demonstration of how various pruning strategies perform before and after lightweight retraining using ∼1.8B tokens. We prune the Nemotron-4 15B model down to the size of Nemotron-3 8B and report the change in distillation loss (KL divergence [28] on logits) and the final LM validation loss with retraining. We see that width (attention, MLP, embedding) pruning outperforms depth, but only after retraining. The last row shows change in loss for the Nemotron-3 8B model.

lion parameters, while the Pythia family [6] offers a selection of eight models with sizes ranging from 80 million to 12 billion parameters. However, training multiple multi-billion parameter models from scratch is extremely time, data and resource-intensive. In this paper, we ask the following question: *can we train one big model, and obtain smaller, more accurate (w.r.t. training from scratch) models from it through a combination of weight pruning and retraining, while only using a small fraction of the original training data?* Achieving such a goal would make producing LLMs targeting different deployment scales significantly cheaper. Weight pruning is a powerful and well-known technique for reducing model size [51, 21]. In this paper, we focus on structured pruning, where blocks of nonzero elements are removed at once from model weights; examples of structured pruning techniques include neuron, attention head, convolutional filter, and depth pruning [32, 18, 53, 4, 34, 55, 26]. While the literature is rich with numerous papers on structured pruning, to an end-user, it's not always clear which technique to use, when, and how to combine them to consistently obtain good pruned models. Pruning is also often accompanied by some amount of *retraining* for accuracy recovery [51]; this phase is extremely expensive in modern LLMs, often requiring access to large amounts of curated data. To the best of our knowledge, no existing work on structured pruning explores data-efficient retraining techniques such as distillation to minimize retraining cost.

In this paper, we perform a thorough empirical exploration of structured pruning and retraining across multiple axes: neurons in feed-forward layers, heads in multi-head attention layers, embedding channels, and model depth. Through our experiments, we gain valuable non-trivial insights on the metrics and hyper-parameters to use for each axis and how to effectively combine axes for higher compression rates. For instance, we discover that pruning neurons and heads alone is initially superior to pruning neurons, heads and embedding channels; however, after a few steps of retraining, this order flips. Similarly, we discover that width pruning works better than depth, but only after some retraining (see Table 1 for a concrete example). We also investigate in detail how a pruned model can be efficiently retrained for optimal performance using minimal additional data. Based on our findings, we develop a practical list of **LLM compression and retraining best practices**. Finally, we apply our findings to prune the Nemotron-4 15B model [43] and produce a family of smaller models, named MINITRON, that compare favorably to similarly-sized models. MINITRON 8B achieves better accuracy than Nemotron-3 8B [39] (using **40×** fewer training tokens) and LLaMa-2 7B [49], and comparable accuracy to Mistral-7B [25], Gemma 7B [48] and Llama-3 8B; likewise, MINITRON 4B outperforms the similarly-sized Gemma2 model and compares favorably to the Phi-2 model.

This paper makes the following key contributions:

1. Provides the first thorough empirical exploration of structured pruning and retraining in LLMs across multiple axes. It offers valuable insights on metrics and hyper-parameter settings for pruning, order of pruning, effects of combining different axes, and retraining techniques focusing on data efficiency.
2. Presents a list of effective and practical *LLM compression and retraining best practices* grounded in extensive empirical evidence.
3. Introduces the MINITRON family of LLMs, which are obtained through direct pruning of the Nemotron-4 15B model. Deriving MINITRON models from Nemotron-4 15B requires up to 40× fewer training tokens compared to training from scratch, while still (1) comparing favorably to various popular community LLMs of similar size, and (2) outperforming state-of-the-art depth and width-pruned models from the literature.

## 2 Pruning Methodology

As shown in Figure 2, we start the pruning process by first computing the importance of each layer, neuron, head, and embedding dimension and then sorting these importance scores to compute a corresponding importance ranking. In this section, we detail how rankings are computed for each axis and then subsequently used to obtain a pruned model.

### 2.1 Background and Notation

We begin with some formal definitions. Multi-Layer Perceptron (MLP) layers have two linear layers with a non-linear activation in between: $\text{MLP}(\mathbf{X}) = \delta\left(\mathbf{X} \cdot \boldsymbol{W}_1^T\right) \cdot \boldsymbol{W}_2$ ; here, $\mathbf{X}$ denotes the input, and $\boldsymbol{W}_1$ and $\boldsymbol{W}_2$ are the two associated weight matrices in the MLP layer. $\boldsymbol{W}_1, \boldsymbol{W}_2 \in \mathbb{R}^{d_{hidden} \times d_{model}}$, where $d_{model}$ and $d_{hidden}$ are the embedding and MLP hidden dimensions, respectively. $\delta(\cdot)$ refers to the non-linear activation function.

We define the Multi-Head Attention (MHA) operation for an input $\mathbf{X}$ as follows: $\text{MHA}(\mathbf{X}) = \text{Concat}(\text{head}_1, ...\text{head}_L) \cdot \boldsymbol{W}^O$, and $\text{head}_i = \text{Attn}(\mathbf{X}\boldsymbol{W}^{Q,i}, \mathbf{X}\boldsymbol{W}^{K,i}, \mathbf{X}\boldsymbol{W}^{V,i})$; here, $\boldsymbol{W}^{Q,i}, \boldsymbol{W}^{K,i}, \boldsymbol{W}^{V,i} \in \mathbb{R}^{d_{head} \times d_{model}}$ and $\boldsymbol{W}^O \in \mathbb{R}^{Ld_{head} \times d_{model}}$ where $d_{head}$ is the size of a single attention head, and $L$ is the total number of heads.

Finally, the Layer Normalization operation (LayerNorm) [5] on an input $\mathbf{X}$ is defined as follows: $LN(\mathbf{X}) = \frac{\mathbf{X}-\mu}{\sqrt{\sigma^2+\epsilon}} \odot \gamma + \beta$, where $\mu$ and $\sigma^2$ represent the mean and variance across the embedding dimensions, $\epsilon$ is a small value for numerical stability, and $\gamma$ and $\beta$ are learnable parameters.

### 2.2 Importance Analysis

Estimating the importance or sensitivity of individual neural network components such as neurons, attention heads, and layers is a well-studied area [9, 13, 41]. In the context of LLMs, recent work has highlighted the ineffectiveness of traditional metrics such as weight magnitude for estimating importance [33]; instead, recent work on structured pruning of LLMs has focused on metrics such as gradient/Taylor [33], cosine similarity [34], and perplexity on a calibration dataset [26].

Owing to their enormous size, computing gradient information on modern LLMs is prohibitively memory and compute-intensive, and one of our primary goals is to avoid this expensive step when trying to obtain importance information. In this paper, we propose a purely *activation-based* importance estimation strategy that simultaneously computes sensitivity information for all the axes we consider (depth, neuron, head, and embedding channel) using a small (1024 samples) calibration dataset and only *forward* propagation passes. We now describe how this strategy is implemented for each individual axis.

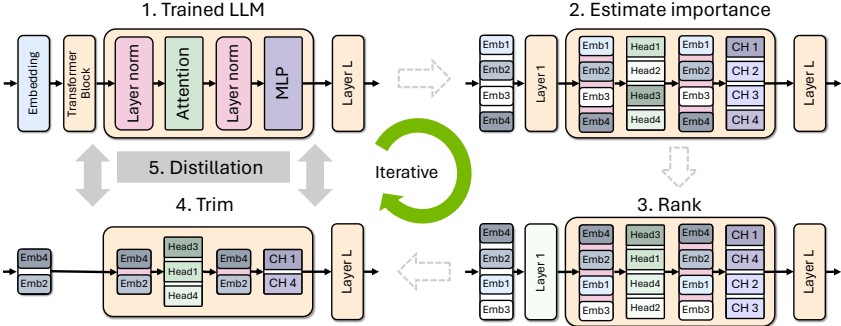

Figure 2: High-level overview of our proposed iterative pruning and distillation approach to train a family of smaller LLMs. On a pretrained LLM, we first evaluate importance of neurons, rank them, trim the least important neurons and distill the knowledge from the original LLM to the pruned model. The original model is replaced with the distilled model for the next iteration of compression.

**Width:** we compute the importance of each head, neuron and embedding channel by examining the activations produced by the *MHA*, *MLP* and *LayerNorm* layers, respectively. We use a small calibration dataset $D$ for this purpose [4]. Formally, we compute activation-based importance scores for heads, neurons, and embedding channels as: $F_{\text{head}}^{(i)} = \sum_{\mathbf{B},\mathbf{S}} \| \text{Attn}(\mathbf{X}\boldsymbol{W}^{Q,i}, \mathbf{X}\boldsymbol{W}^{K,i}, \mathbf{X}\boldsymbol{W}^{V,i})\|_2$, $F_{\text{neuron}}^{(i)} = \sum_{\mathbf{B},\mathbf{S}} \mathbf{X}\left(\boldsymbol{W}_1^i\right)^T$, and $F_{\text{emb}}^{(i)} = \sum_{\mathbf{B},\mathbf{S}} LN(\mathbf{X})_i$. Here, $\boldsymbol{W}_1^i$ refers to the $i^{\text{th}}$ row of the weight matrix $\boldsymbol{W_1}$. $\sum_{\mathbf{B},\mathbf{S}}$ refers to aggregation along the batch and sequence dimensions. We observe from our experiments that performing a simple summation here is not always optimal. To this end, we perform a detailed evaluation of various aggregation functions along each of these dimensions and their corresponding performance in Table 11. Specifically, for a sequence of scores $\mathbf{S}$, we try three functions: (1) mean(abs): $\frac{1}{n}\sum_{i=1}^n |\mathbf{S}_i|$ (hereafter referred to as just *mean*), (2) L2 norm: $\sqrt{\sum_{i=1}^n \mathbf{S}_i^2}$, and (3) variance: $\frac{1}{n}\sum_{i=1}^n (\mathbf{S}_i - \bar{\mathbf{S}})^2$. Layer-wise scores are then summed up to obtain network-wide importance scores for each axis.

**Depth (Layers):** for depth pruning, we evaluate the importance of each layer using two metrics: (1) perplexity (PPL) [26] and (2) Block Importance (BI) [34]. For PPL-based ranking, we simply remove a single layer and compute its effect on perplexity of this pruned model; this serves as the "importance" or sensitivity of the layer [26]. BI [34] uses the cosine distance between the input and output of a layer to estimate layer sensitivity. The BI score of layer $i$ is computed as: $\text{BI}_i = 1 - \mathbb{E}_{X,t} \frac{\mathbf{X}_{i,t}^T \mathbf{X}_{i+1,t}}{\|\mathbf{X}_{i,t}\|_2 \|\mathbf{X}_{i+1,t}\|_2}$, where $\mathbf{X}_i$ refers to the input to layer $i$, and $\mathbf{X}_{i,t}$ denotes the $t^{th}$ row of $\mathbf{X}_i$. The BI of all layers can be computed in a single forward pass, giving it a significant speed advantage over PPL-based importance. Additionally, following Gromov et al. [14], we can extend BI to estimate importance of several contiguous layers at the same time.

**Iterative Importance:** in this setting, we iteratively alternate between pruning and importance estimation for a given axis or combination of axes. Formally, given number of iterations $T$ and source and target dimensions (layers, heads, etc.) $d_s$ and $d_t$, respectively, we iteratively compute importance on $d_s - i \cdot \left(\frac{d_s - d_t}{T}\right)$ dimensions and prune to $d_s - (i+1) \cdot \left(\frac{d_s - d_t}{T}\right)$ dimensions; $i \in [0, T-1]$. We evaluate the effectiveness of iterative importance estimation in Table 12.

### 2.3 Obtaining a Pruned Model

Figure 2 provides an overview of how pruned models are obtained. For a given architecture configuration, we first rank the elements of each axis according to the computed importance and perform trimming (reshaping) of the corresponding weight matrices directly. For neuron and head pruning, we trim MLP and MHA layer weights, respectively. In the case of embedding channels, we trim the embedding dimension of the weight matrices in MLP, MHA, and LayerNorm layers.

When pruning attention heads, we add the *residual* info from the pruned heads back into the remaining heads, with the aim of preserving relevant knowledge from the pruned heads. This idea is an MHA analog of Layer Collapse [55] for depth pruning and provides a boost to model accuracy in our experiments. Formally, given $L$ original attention heads $head_1, head_2, ..., head_L$ being pruned to $K$ heads, each new head will have the form (for the $i^{th}$ head): $head_i + (head_i - head_{2K-i+1})$ for $i \in [K - (L - K), K]$. In case of grouped query attention [3], we apply this strategy only to the query heads.

**Lightweight Neural Architecture Search:** Figure 3 provides an overview of our search strategy for finding optimal architecture configurations. Given a search space and parameter budget (left side of the figure), we enumerate all feasible architectures meeting the parameter budget. At this stage, while it's possible to further reduce the search space size using strategies such as genetic search and/or Bayesian optimization, we found that sticking to commonly used neuron, head and embedding dimensions, along with a reasonably narrow target parameter range (less than 1 billion) was sufficient to obtain tractable solution sets (less than 20 candidates). The feasible candidates then undergo **lightweight retraining** ($\sim$1.8B tokens in this work). We show in Figure 9 that this retraining stage stabilizes relative rankings and helps us find a more accurate candidate to train further. We note that parameter-efficient fine-tuning techniques such as LoRA [23] can also be applied at this stage; we leave the exploration of such techniques to future work.

---

[4]We provide additional details of the calibration dataset in Section 4.

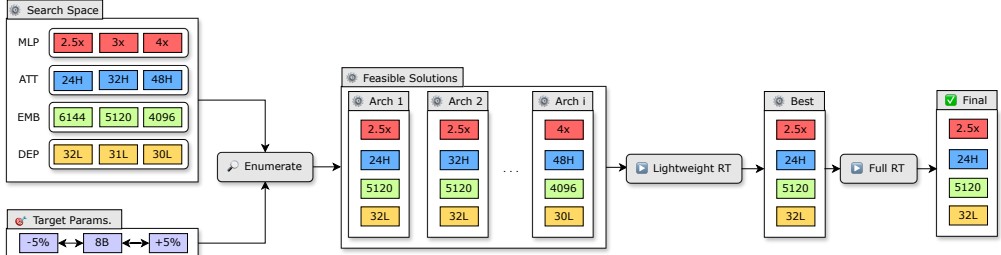

Figure 3: Overview of our neural architecture search algorithm. We perform a search on multiple axes: number of layers, attention head count, MLP and embedding dimensions to arrive at a set of feasible architectures meeting the parameter budget. RT refers to retraining.

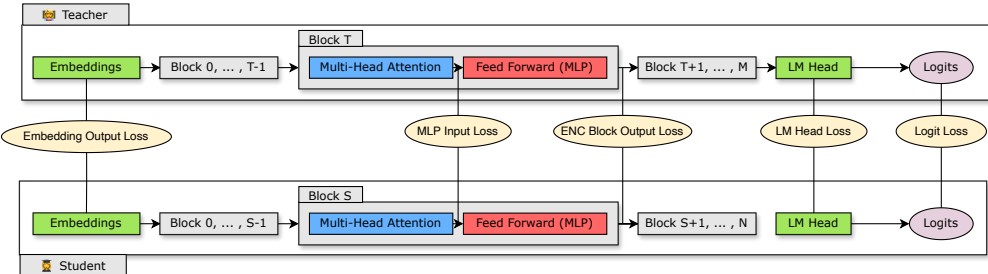

Figure 4: Overview of Distillation. A *student* model with $N$ layers is distilled from a *teacher* model with $M$ layers. The *student* learns by minimizing a combination of embedding output loss, logit loss and transformer encoder specific losses mapped across *student* block $S$ and *teacher* block $T$.

## 3 Retraining

We use the term *retraining* to refer to the accuracy recovery process following pruning. In this paper, we explore two retraining strategies: (1) conventional training, leveraging ground truth labels, and (2) knowledge distillation using supervision from the unpruned model (teacher).

**Retraining with Knowledge Distillation:** Knowledge Distillation (KD) involves transfer of knowledge from a larger or more complex model called the teacher to a smaller/simpler model called the student [20]. The knowledge transfer is achieved by having the student model mimic the output and/or the intermediate states of the teacher model. In our case, the the uncompressed and pruned models correspond to the teacher and student, respectively.

The output probability distribution of an LLM for a given token $x_i$ is computed as: $p(x_i, \tau) = \frac{\exp\left(\frac{x_i}{\tau}\right)}{\sum_{j=1}^{|V|} \exp\left(\frac{x_j}{\tau}\right)}$, where $\tau$ is the softmax temperature and $|V|$ is the vocabulary size. Logit-based KD loss across the sequence of all output tokens is represented as: $L_{\text{logits}} = \frac{1}{l} \sum_{k=1}^{l} \text{Loss}(p_t^k(x, \tau), p_s^k(x, \tau))$; here, $p_t^k(x, \tau)$ and $p_s^k(x, \tau)$ represent the teacher and student probability distributions on the $k^{th}$ token, respectively, and $l$ represents the sequence length.

For distillation, we explore various loss functions, and several combinations of intermediate states and mappings across the Transformer model as the loss components, along with their respective trade-offs. This is illustrated in Figure 4. The intermediate state-based KD loss across a sequence of Transformer-specific hidden states is represented as: $L_{is} = \frac{1}{l} \sum_{k \in H} \sum_{i=1}^{l} Loss_k(h_t^{ki}, h_s^{ki})$, where $h_t^{ki}$ and $h_s^{ki}$ represent the $k^{th}$ teacher and student hidden state for the $i^{th}$ token, respectively, and $l$ represents the sequence length; $H$ is the set of chosen intermediate states. The mismatch in student and teacher hidden states is handled by learning a shared linear transformation during distillation to upscale the student hidden state to the teacher hidden state dimension. The hidden states used are always post LayerNorm. We report our experimental results for retraining in Section 4.3.

| | Benchmark | Metric | Llama-3 | Llama-2 | Mistral | Gemma | Nemotron-4 | Nemotron-3 | MINITRON |
|---|---|---|---|---|---|---|---|---|---|
| | # Parameters | | 8B | 6.7B | 7.3B | 8.5B | 15.6B | 8.5B | 8.3B |
| | # Non-Emb. Params | | 5.9B | 6.4B | 7B | 7.7B | 12.5B | 6.4B | 6.2B |
| | # Training Tokens | | >15T | 2T | 8T | 6T | 8T | 3.8T | **94B** |
| Knowledge,Logic | winogrande (5) | acc | 77.6 | 74 | 78.5 | 78 | 83.6 | 75.9 | **79.0** |
| | arc_challenge (25) | acc_norm | 57.8 | 53 | 60.3 | **61** | 58.8 | 52.8 | 52.6 |
| | MMLU(5) | acc | **65.3** | 46 | 64.1 | 64 | 66.6 | 54.7 | 63.8 |
| | hellaswag(10) | acc_norm | 82.1 | 79 | **83.2** | 82 | 84.6 | 78.5 | 80.7 |
| | gsm8k(5) | acc | 50.3 | 14 | 37 | 50 | 48.5 | 24.0 | **51.3** |
| | truthfulqa(0) | mc2 | 43.9 | 39 | 42.6 | **45** | 40.7 | 36.5 | 42.6 |
| | XLSum en (20)(3) | rougeL | 30.9 | 31 | 4.80 | 17 | 32 | 30.9 | **31.2** |
| Coding | MBPP(0) | pass@1 | **42.4** | 20 | 38.8 | 39 | 38 | 27.04 | 35.2 |
| | humaneval (n=20)(0) | pass@1 | 28.1 | 12 | 28.7 | **32** | 35.4 | 20.7 | 31.6 |

Table 2: Performance of our pruned MINITRON 8B model compared to multiple baselines: the original Nemotron-4 15B, the previous generation Nemotron-3 8B, and multiple community models. MINITRON 8B uses 40× fewer tokens than Nemotron-3 8B. All evaluations run by us, except for entries marked with *, which we report from the corresponding papers.

| | Benchmark | Metric | Phi-2 | Gemma | Gemma2* | Qwen2* | MiniCPM* | MINITRON |
|---|---|---|---|---|---|---|---|---|
| # Parameters | | | 2.7B | 2.5B | 2.6B | 1.5B | 2.7B | 4.2B |
| # Non-Emb. Params | | | 2.5B | 2B | 2B | 1.3B | 2.4B | 2.6B |
| # Training Tokens | | | 1.4T | 3T | 2T | 7T | 1.1T | **94B** |
| Knowledge, Logic | winogrande (5) | acc | **74** | 67 | 70.9 | 66.2 | - | **74.0** |
| | arc_challenge (25) | acc_norm | **61** | 48 | 55.4 | 43.9 | - | 50.9 |
| | MMLU(5) | acc | 57.5 | 42 | 51.3 | 56.5 | 53.5 | **58.6** |
| | hellaswag(10) | acc_norm | **75.2** | 72 | 73.0 | 66.6 | 68.3 | 75.0 |
| | gsm8k(5) | acc | 55 | 18 | 23.9 | **58.5** | 53.8 | 24.1 |
| | truthfulqa(0) | mc2 | 44 | 33 | - | **45.9** | - | 42.9 |
| | XLSum en (20)(3) | rougeL | 1 | 11 | - | - | - | **29.5** |
| Coding | MBPP(0) | pass@1 | **47** | 29 | 29.6 | 37.4 | - | 28.2 |
| | humaneval (n=20)(0) | pass@1 | **50** | 24 | 17.7 | 31.1 | - | 23.3 |

Table 3: Performance of MINITRON 4B model compared to similarly-sized community models. All evaluations run by us, except for entries marked with *, which we report from the corresponding papers. We only compare to base models without SFT and DPO, therefore Phi-3 is excluded.

The total loss $L$ is computed as $L = L_{\text{CLM}} + L_{logits} + \alpha \times L_{is}$; where $L_{CLM}$ is the student cross-entropy loss against the ground truth labels, and $\alpha$ is a weighting coefficient. As the magnitudes of $L_{logits}$ and $L_{is}$ differ significantly, we found that computing $\alpha$ dynamically as $\frac{L_{logits}}{L_{is}}$ achieves better results compared to using a constant.

## 4 Experiments and Results

We evaluate our pruning strategy on the Nemotron-4 family of models [43]; specifically, we compress the Nemotron-4 15B model with 15.6 billion parameters down to two target parameter ranges: (1) 8 billion, and (2) 4 billion. We use the NVIDIA Megatron-LM framework [47] to implement our pruning and distillation algorithms for compression and retraining.

**Data and Training Hyperparameters:** we use the Nemotron-4 curated 8 trillion token (8T) base pretraining dataset and the continued training dataset (CT) [42, 44, 43]. We use the 8T training blend for all our ablations and use a combination of both data blends to retrain our final models. Unless otherwise specified, we use 1.8 billion tokens (400 steps) for lightweight retraining. The calibration dataset $D$ used for importance estimation consists of 1024 samples drawn randomly from the full dataset. We use the same optimizer settings and data split as [43] with cosine LR decay schedule from $2^{-4}$ to $4.5^{-7}$.

**Downstream Tasks:** following Touvron et al. [49], we evaluate our models of similar size on a series of downstream tasks, including MMLU [19], HumanEval [8] for Python code generation, several question-answering datasets for common-sense reasoning: Arc-C [10], HellaSwag [56], TruthfulQA [29] and WinoGrande [45] and XL-Sum English [17] for summarization. We report the 5-shot performance on MMLU, 5-shot on Winogrande, 25-shot on ARC-Challenge, 10-shot

on HellaSwag, 0-shot on 20% of XL-Sum and average pass@1 scores for HumanEval and MBPP. For pass@1 scores we use a temperature of 0.2 and nucleus sampling [22] with top-p = 0.95. For instruction-tuned models, we use MT-Bench [57], Instruction-Following Eval (IFEval) [59], ChatRAG-Bench [30], and Berkeley Function Calling Leaderboard (BFCL) [54].

## 4.1 Main Pruning Results

We start by introducing the following list of **structured compression best practices**:

> 1. To train a family of LLMs, train the largest one and prune+distill iteratively to smaller LLMs.
> 2. Use (`batch=L2, seq=mean`) importance estimation for width axes and PPL/BI for depth.
> 3. Use single-shot importance estimation; iterative provides no benefit.
> 4. Prefer width pruning over depth for the model scales we consider ($\leq$ 15B).
> 5. Retrain exclusively with distillation loss using KLD instead of conventional training.
> 6. Use (logit+intermediate state+embedding) distillation when depth is reduced significantly.
> 7. Use logit-only distillation when depth isn't reduced significantly.
> 8. Prune a model closest to the target size.
> 9. Perform lightweight retraining to stabilize the rankings of searched pruned candidates.
> 10. If the largest model is trained using a multi-phase training strategy, it is best to prune and retrain the model obtained from the final stage of training.

We arrive at this list through a detailed set of ablations and experiments, and each point is backed by empirical evidence, as we demonstrate in the rest of this section and the Appendix. We use this list to obtain our MINITRON pruned and retrained models, whose performance is shown in Tables 2 and 3. Here, we compare the performance of our pruned models to multiple baselines: (1) the original Nemotron-4 15B model, (2) the previous generation Nemotron-3 8B model, and (3) a set of similarly-sized community models, all trained from scratch with trillions of tokens. Evaluation is performed on the downstream tasks described earlier in this Section. In both tables, we list the number of full and non-embedding parameters, along with the number of training tokens used to arrive at the model.

We further compare the MINITRON models to state-of-the-art depth and width-pruned baselines in Table 4; namely, LLM-Pruner [33], SliceGPT [4], LaCo [55], ShortGPT [34], and Sheared LLaMa [53]. Table 10 (Appendix) lists the architecture details of the Nemotron and MINITRON models shown in Tables 2 and 3. In the following subsections, we will go into more detail on how we arrived at the MINITRON pruned models.

From Table 2, we notice that MINITRON 8B compares favorably to the latest community models of the same size. Specifically, we outperform Nemotron-3 8B and LLaMa-2 7B, and perform on par with Mistral 7B, Gemma 7B and LLaMa-3 8B, all while using significantly fewer training tokens. MINITRON 8B also significantly outperforms multiple depth-pruned models of larger size ($\sim$ 10B parameters) (Table 4). From Table 3, we notice that our smaller model, MINITRON 4B, **retains model capabilities better** compared to small specialized models that score highly only on some tasks, outperforms the Gemma2 model and is significantly superior to multiple depth and/or width pruned models shown in Table 4.

**Instruction Tuning:** to better understand how MINITRON models perform after supervised fine-tuning (SFT), we perform SFT on MINITRON 4B using instruction-tuning data used for Nemotron-4 340B [38] to create MINITRON 4B-instruct, and evaluate it on various tasks, including instruction-following and roleplay (IFEval and MT-Bench), RAG QA (ChatRAG-Bench), and function calling (BFCL). The results for this experiment are shown in Tables 5 to 8. Tables 5 to 7 demonstrate that MINITRON 4B-instruct has strong instruction-following, roleplay and RAG capabilities, beating similarly sized models across all tasks. On function calling (Table 8), MINITRON 4B-instruct outperforms Gemma-2B-IT and even Llama-3-8B-instruct.

**Best Practice #1:** in summary, Tables 2 - 8 provide strong empirical evidence to support the claim that training one single big model, and obtaining smaller ones from it through pruning + retraining achieves higher accuracy and is extremely cost/compute-efficient when compared to training them from scratch. Further, our efficient retraining strategy also **eliminates the need to curate trillions of tokens of data**.

| | | Models | | | | | |
|---|---|---|---|---|---|---|---|
| **Benchmark** | **Metric** | **LLMPruner** | **SliceGPT** | **LaCo** | **ShortGPT** | **Sheared LLaMa** | **MINITRON** |
| **8 Billion** | | | | | | | |
| # Parameters | | 9.8B | 9.9B | 9.8B | 9.8B | - | 8.3B |
| # Non-Emb. Params | | 9.5B | 9.5B | 9.5B | 9.5B | - | 6.2B |
| MMLU(5) | acc | 25.2 | 37.1 | 45.9 | 54.7 | - | **63.8** |
| hellaswag(10) | acc_norm | 67.8 | 55.7 | 64.4 | 66.6 | - | **80.7** |
| **4 Billion** | | | | | | | |
| # Parameters | | 4.8B | 4.9B | 4.9B | 4.9B | 2.7B | 4.2B |
| # Non-Emb. Params | | 4.5B | 4.6B | 4.6B | 4.6B | 2.5B | 2.6B |
| winogrande (5) | acc | - | - | - | - | 64.2 | **74** |
| arc_challenge (25) | acc_norm | - | - | - | - | 41.2 | **50.9** |
| MMLU(5) | acc | 23.33 | 28.92 | 26.45 | 43.96 | 26.4 | **58.6** |
| hellaswag(10) | acc_norm | 56.46 | 50.27 | 55.69 | 53.02 | 70.8 | **75** |
| gsm8k(5) | acc | - | - | - | - | 23.96 | **24.1** |

Table 4: Performance of MINITRON models w.r.t recent state-of-the-art models obtained through depth/width pruning. Top and bottom halves show results for MINITRON 8B and 4B, respectively.

| Model | Non-Emb. Params | Tokens | Total |
|---|---|---|---|
| MINITRON 4B-instruct | 2.6B | **90B** | **6.46** |
| Phi-2 | 2.5B | 1.4T | 4.29 |
| Qwen-1.5 Chat | 1.2B | N/A | 5.29 |
| Gemma-2B-IT | 2B | 6T | 5.19 |
| StableLM 2 Chat | 1.6B | 2T | 5.42 |
| TinyLlama v1.0 Chat | 1.1B | 3T | 3.46 |

Table 5: Evaluation results on MT-Bench.

| Model | Prompt-level Acc. (strict) | Prompt-level Acc. (loose) | Instruction-level Acc. (loose) |
|---|---|---|---|
| MINITRON 4B-instruct | **68.76** | **73.01** | **81.29** |
| Gemma-2B-IT | - | 28.70 | 40.50 |
| Qwen2-1.5B-Instruct | 29 | - | - |

Table 6: Evaluation results on IFEval.

**Cost Savings for Training a Model Family:** the FLOPs required per training step [5] for the 15B, 8B, and 4B models in the Nemotron-4 model family are, respectively: 4.4e17, 2.5e17 and 1.2e17. With the assumption that each model in the family is trained with an equivalent token count, steps and batch size, we obtain the following FLOP count for training each model in the family from scratch: $(4.4e17 + 2.5e17 + 1.2e17) \times$ steps. As noted from Tables 2 and 3, our approach requires $40\times$ fewer training tokens for each additional model, hence resulting in the following updated FLOP count for the family: $(4.4e17 + 2.5e17/40 + 1.2e17/40) \times$ steps; the corresponding cost savings for training the full Nemotron-4 family using our approach is thus $1.8\times$.

We now dive deeper into our empirical ablations that help us arrive at the list of best practices. Unless otherwise specified, we run these ablations on the Nemotron-4 15B checkpoint prior to continued training with the CT data blend.

## 4.2 Obtaining the Best Pruned Model

**Best Aggregation Metric (Best Practice #2):** we start by exploring the best aggregation metric for use with our activation-based pruning criteria (see Section 2.2 for more details). Table 11 shows how zero-shot LM loss and Wikitext2 perplexity [35] vary w.r.t different intra-batch and sequence aggregation functions. Here, the Nemotron-4 15B model is pruned to the Nemotron-3 8B architecture with no retraining. We notice that there is significant variation in zero-shot performance based on the aggregation metric, indicating the importance of selecting the right one. Both (`batch=L2`, `seq=mean`) and (`mean, mean`) perform well; in the remainder of the paper, we use (`l2, mean`) primarily due to its slightly better performance on the 8T dataset. To further evaluate if these relative rankings hold after retraining, we perform a related experiment: we prune the same 15B model to 8B using: (1) the best ((`L2, mean`) metric, and (2) a poorly performing (`L2, L2`) metric, and perform retraining on both for 400 steps (~1.8B tokens). The results of this experiment are shown in Figure 5. From the Figure, we conclude that these rankings continue to hold post-retraining.

**Iterative Importance (Best Practice #3):** we evaluate whether iterative importance estimation provides any benefit (described in Section 2.2) and report results in Table 12. Here, we take the Nemotron-4 15B model and prune the embedding dimension alone using number of iterations T=1, 2, and 4 iterations to the target value of 4096. We then perform lightweight retraining of all 3 candidates

---

[5]Assume a batch size of 1152.

| Model | Avg |
|---|---|
| MINITRON 4B-instruct | **41.11** |
| Gemma-2B-IT | 33.31 |

Table 7: Evaluation results on ChatRAG-Bench.

| Model | Avg. |
|---|---|
| MINITRON 4B-instruct | **53.09** |
| Gemma-2B-IT | 41.63 |
| Llama-3-8B-instruct | 50.51 |

Table 8: Evaluation results on BFCL v2.

for 1.8B tokens. From the Table, we observe that while the iterative approach appears to be better before retraining, all 3 candidates converge to the same loss value, indicating no benefit.

**Combining Depth and Width (Best Practice #4):** we perform a simple experiment to compare the efficacy of width vs. depth pruning. Using the PPL and BI metrics defined in Section 2.2, we remove the 16 least important layers from the Nemotron 15B model based on both metrics to arrive at two variants of depth pruned models. We also perform neuron, head and embedding channel pruning to target the Nemotron-3 8B model and arrive at the width pruned variant. Finally, we combine depth (remove 4 least important layers) and width pruning to arrive at the fourth variant. We report the results of this experiment in Table 13. We notice that even though the depth-width pruned variant has a lower loss post-pruning, we see the results flip around 200 steps of retraining (0.8B tokens); Table 1 and Figure 6 further illustrate this point.

## 4.3 Retraining and Search

**Distillation vs. Conventional Training (Best Practice #5):** in this experiment, we compare: (1) training a 4B model with random initialization (4B-Random-Init), (2) pruning 15B to 4B, followed by retraining with conventional training (4B-Pruned), and (3) pruning 15B to 4B, and then retraining with distillation using the 15B model as the teacher (4B-Pruned-Distill). Since distillation adds training overheads (additional forward pass on the teacher model), we compare approaches under iso-compute settings. Table 14 shows the results. Here, we observe a significant improvement in MMLU for (3), while both (1) and (2) score randomly. On HellaSwag, (3) > (2) > (1). This clearly demonstrates the superiority of distillation over conventional training after pruning.

**Choice of Loss Function (Best Practice #5):** we experiment with Kullback-Leibler divergence (KLD), MSE, cosine similarity and reverse KLD (R-KLD) to compute $L_{logits}$. Recent work has shown R-KLD [15, 27] to be a better fit than KLD in the SFT/instruction-following setting, and Agarwal et al. [2] claim the choice of loss is task-dependent. We observe from Table 15 and 16 that **KLD** is the best choice for pruned base model training.

**Choice of Losses (Best Practices #6 and #7):** typically, a weighted combination of $L_{CLM}$ and $L_{logits}$ is used for distillation. We find that using $L_{logits}$ alone results in the best performance as shown in Table 15. For $L_{is} = L_{emb} + L_{att} + L_i + L_o$ , we make several observations similar to Lu et al. [31]; these are listed in Appendix A.6 and in Table 17. Most notably, we observe no improvements from using $L_{is}$ when retraining models that **don't prune the depth axis significantly**, such as MINITRON 8B and MINITRON 4B and hence use $L_{logits}$ alone in such cases (see Table 18).

**One-shot vs Iterative Pruning and Distillation Across Model Sizes (Best Practice #8):** compressing Nemotron-4 15B to MINITRON 4B requires an aggressive 73.3% reduction of original model weights. We hypothesize that aggressive one-shot pruning loses out on important capabilities of the base LLM. We thus explore a simple iterative two-step pruning and retraining strategy where we first prune and retrain Nemotron-4 15B to create MINITRON 8B ($\sim$46% reduction) and further prune and retrain the latter to MINITRON 4B ($\sim$50% reduction). Table 14 (last two rows) shows the comparison between single-shot and iterative pruning, and demonstrates that iterative achieves a 12% improvement in the MMLU scores compared to the one-shot strategy. During the final retraining step, we observe that using Nemotron-4 15B as the teacher achieves superior results compared to using MINITRON 8B. We provide additional ablations on one-shot vs. iterative pruning in Appendix A.7.

**Search with Retraining (Best Practice #9):** for lightweight neural architecture search, we use the search spaces defined in Table 9 for MINITRON 8B and 4B. We further specify a target parameter range of 8 and 4 billion parameters for the respective models, with a tolerance of 5%. With these settings, we obtain 15 and 18 feasible candidates for the 8B and 4B parameter targets, respectively. The architecture configurations for these candidates are provided in Table 19. As described in Section 2.3, we perform lightweight retraining of all feasible candidates. Figure 9 illustrates how

validation loss changes for the 8B candidates as training progresses. We notice that relative rankings undergo significant changes up to $\sim 300$ steps, and then stabilize.

| Target | Layers | Heads | MLP Exp. Factor | Embedding |
|---|---|---|---|---|
| MINITRON 8B | [29-32] | {32,48} | {2.5,3,3.5,4} | {4096,4680,5120,5632,6144} |
| MINITRON 4B | [29-32] | {24,32,48} | {2.5,3,3.5,4} | {2560,3072,3584,4096,4608} |

Table 9: MINITRON 8B and 4B search space.

**Single vs Multi-Phase Retraining (Best Practice #10):** Recent studies [1] [24] [43] [46] have shown improved results with multi-phase pretraining routines. Initially, models are trained on web data, followed by a lightweight phase with cleaner data. We explored two compression techniques: (1) prune the phase 1 checkpoint, retrain with portions of phase 1 and 2 data, and (2) prune the phase 2 checkpoint, retrain with a portion of phase 2 data. Table 20 shows that (2) is sufficient to regain accuracy and surpasses (1). This strategy is used for our best models, also suggesting that for further aligned models, it may suffice to prune the aligned model and retrain with a portion of the alignment dataset.

## 5 Related Work

**Structured LLM Pruning:** there have been a number of recent structured pruning papers specifically targeting LLMs; we can broadly classify these works into two main categories: (1) ones that prune only depth (layers), (2) ones that prune width (attention heads, MLP intermediate dimension, etc.) and/or depth. Recent work in the first category (depth pruning) includes ShortGPT [34], LaCo [55], and Shortened LLaMa [26]; for pruning layers in MINITRON models, we reuse and extend the metrics proposed in some of these works (eg: block importance from ShortGPT [34]). A number of recent papers have also proposed new saliency metrics and pruning strategies targeting width dimensions: namely, embedding channels, attention heads, and MLP intermediate channels [11, 4, 53, 33]. Most work in this category uses learnable masks, combined with an Augmented Lagrangian loss formulation to arrive at optimal width masks [4, 53, 33]. At LLM scale, this strategy has multiple disadvantages: (1) it requires compute and memory-intensive gradient computations, and (2) it requires a considerable amount of data and fine-tuning to arrive at reasonable masks. The notable exception in this line of work is Dery et al. [11], which recognizes these limitations and proposes saliency metrics that can be computed with only forward passes. To the best of our knowledge, we provide the first pruning strategy that (1) simultaneously targets both width and depth dimensions, (2) works at LLM scale (i.e., uses only forward passes for computing importance and uses a small fraction of pretraining data), and (3) achieves state-of-the-art compression and accuracy.

**Post-pruning Accuracy Recovery:** recent work has leveraged either a teacher model which is larger/better [2, 27] or teacher-generated synthetic data [1, 16, 36, 37] to improve the accuracy of an existing trained smaller base model in the Supervised Fine Tuning (SFT)/instruction following setting. Compared to recent width and depth pruning work [26, 34, 53], to the best of our knowledge, we are the first to employ distillation from an uncompressed teacher to improve the retraining of structurally-pruned student models.

## 6 Conclusions

This paper has presented a thorough empirical exploration of structured pruning and retraining in LLMs, offering unique insights into pruning order, effects of combining pruning axes, and retraining techniques for minimal data use. We have developed a set of compression and retraining best practices, backed by extensive empirical evidence, which we employ to prune the Nemotron-4 15B model by a factor of 2-4$\times$. Our compressed MINITRON models are significantly cheaper to obtain compared to training each model from scratch (requiring up to 40$\times$ fewer training tokens), while still performing favorably to a number of similarly-sized community models; MINITRON models also outperform multiple state-of-the-art depth and width pruned models from the literature. **Limitations:** one notable limitation of our work is that we currently apply our proposed techniques only on the Nemotron family of models; we plan to address this by pruning other model families in future work. Also, even though it is short, our method requires full model retraining.

## Acknowledgments and Disclosure of Funding

We would like to thank Ameya Sunil Mahabaleshwarkar, Hayley Ross, Brandon Rowlett, Oluwatobi Olabiyi, Ao Tang, and Yoshi Suhara for help with producing the instruction-tuned versions of MINITRON; additionally, James Shen for TRT-LLM support, and Sanjeev Satheesh, Oleksii Kuchaiev, Shengyang Sun, Jiaqi Zeng, Zhilin Wang, Yi Dong, Zihan Liu, Rajarshi Roy, Wei Ping, and Makesh Narsimhan Sreedhar for help with datasets. We'd also like to gratefully acknowledge the insightful discussion and feedback from Chenhan Yu and Daniel Korzekwa.

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

# A Appendix

## A.1 Pruned Architecture Details

| Model | Layers | Hidden Size | Att. Heads | Query Groups | MLP Hidden | Parameters |
|---|---|---|---|---|---|---|
| Nemotron-4 15B | 32 | 6144 | 48 | 8 | 24576 | 15.6B |
| Nemotron-3 8B | 32 | 4096 | 32 | 32 | 16384 | 8.5B |
| MINITRON 8B | 32 | 4096 | 48 | 8 | 16384 | 8.27B |
| MINITRON 4B | 32 | 3072 | 24 | 8 | 9216 | 4.19B |

Table 10: Architecture details of the uncompressed Nemotron and pruned MINITRON models. Vocabulary size is 256k for all models.

## A.2 Width Pruning

**Best Aggregation Metric for Width Pruning:** Results post-pruning (zero-shot) are shown in Table 11 and after lightweight retraining in Figure 5.

| Batch | Sequence | 8T LM Loss | WikiText2 LM Loss |
|---|---|---|---|
| L2 | L2 | 8.73 | 8.37 |
| **L2** | **mean** | **7.18** | **7.23** |
| L2 | var | 8.18 | 8.61 |
| mean | L2 | 8.41 | 7.84 |
| **mean** | **mean** | **7.21** | **6.89** |
| mean | var | 7.94 | 8.29 |
| var | L2 | 9.01 | 9.30 |
| var | mean | 8.34 | 8.72 |
| var | var | 10.55 | 11.14 |

Table 11: Zero-shot performance of activation-based importance with different batch and sequence aggregation metrics. LM loss is reported on the validation set of the 8T and WikiText2 datasets.

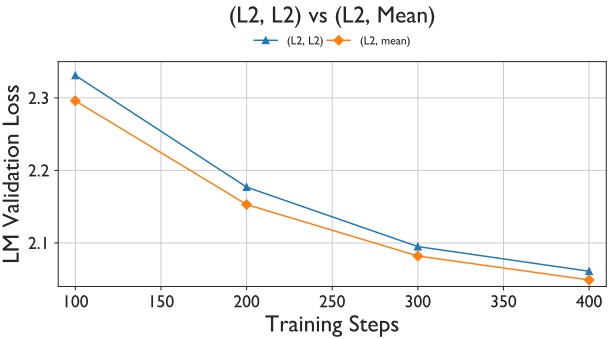

Figure 5: LM validation loss curve for retraining of two pruned candidates with (L2, L2) and (L2, Mean) metrics for (batch, sequence) aggregation strategies.

| Iterations | Initial (Zero-Shot) Validation Loss | Final Validation Loss |
|---|---|---|
| T=1 | 5.43 | 1.92 |
| T=2 | 5.55 | 1.92 |
| T=4 | **5.24** | 1.92 |

Table 12: Comparison of one-shot importance estimation and pruning vs iterative importance estimation and pruning the embedding dimension from the original size to the target size. LM validation loss is reported before and after lightweight retraining.

## A.3 Depth vs. Width Pruning

| Model | Parameters | LM Loss |
|---|---|---|
| MINITRON 8B Depth (PPL) [26] | 9.39B | 2.155 |
| MINITRON 8B Depth (BI) [34] | 9.39B | 2.177 |
| **MINITRON 8B Width** | **7.74B** | **2.049** |
| MINITRON 8B Depth + Width | 7.91B | 2.062 |

Table 13: Comparison of retraining LM loss across different pruning strategies post retraining with 1.8B tokens. We explore depth only, width only, and a combination of both. Width only strategy though with the least parameter count outperforms the rest.

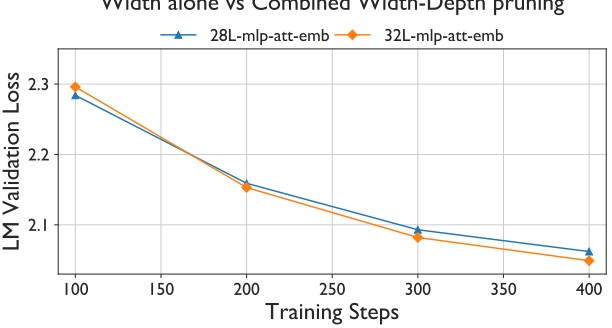

Figure 6: Comparison of retraining LM validation loss curves across two pruning choices, width alone vs combined depth and width. We observe a flip in the ranking prior to 200 steps of retraining, showcasing the need for a lightweight retraining phase.

## A.4 Distillation vs. Conventional Training

| Model | Tokens | Hellaswag | MMLU |
|---|---|---|---|
| 4B-Random-Init | 150B* | 46.22 | 24.36 |
| 4B-Random-Init | 400B | 48.23 | 26.24 |
| 4B-Pruned (prune Nemotron-4 15B) | 150B* | 50.85 | 24.57 |
| **4B-Pruned-Distill (prune Nemotron-4 15B)** | **100B*** | **51.04** | **37.81** |
| **4B-Pruned-Distill (prune MINITRON 8B)** | **100B*** | **52.04** | **42.45** |

Table 14: Accuracy comparison across different strategies to train a 4B model. Pruning the 15B model and distillation results in a gain of 4.8% on Hellaswag and 13.5% on MMLU compared to training from scratch with equivalent compute. Pruning an 8B model instead of a 15B model results in an additional gain of 1% and 4.6% on the benchmarks. * Indicates settings with iso-compute.

## A.5 Retraining with Distillation

**Choice of loss function:** In our experiments with the previous generation of Nemotron models in Table 15, we see that KLD consistently outperforms R-KLD, cosine and MSE. WSL-KD [58] also performs inferior to KLD. Hence, we do not repeat all these studies with the experiment setup in Section 4, rather only a subset as shown in Table 16.

**Temperature:** We experiment with $\tau$ =0.1, 0.5, 1.0, 3.0 in the softmax computation. Literature shows vision (classification) models output a spikey logit distribution and softening the logit distribution with temperature > 1 results in an improvement when using distillation. However, LLM logit distributions have higher entropy and hence the inspiration for temperature < 1 to reduce the noise. We observe best results when $\tau$**=1.0**.

| Loss | LM loss | WikiText PPL |
|---|---|---|
| $L_{CLM} + L_{logits}$(MSE) | 2.144 | 9.007 |
| $L_{CLM} + L_{logits}$(RKLD) | 2.140 | 9.008 |
| $L_{CLM} + L_{logits}$(Cosine) | 2.134 | 8.965 |
| $L_{CLM} + L_{logits}$(**KLD**) | **2.117** | **8.791** |
| $L_{logits}$(**KLD**) | **2.107** | **8.720** |

Table 15: LM loss comparison for various loss functions and loss components on Nemotron-3 8B. The loss component Llogits alone with forward KLD loss outperforms the rest.

| Loss Function | LM loss |
|---|---|
| $L_{logits}$(RKLD) | 2.665 |
| $L_{logits}$(**KLD**) | **2.155** |

Table 16: Comparison of loss functions with MINITRON 8B-Depth-pruned. LM loss is reported on the validation sets of the 8T.

**Top-K:** Inspired by the top-K/top-P sampling approach used in LLM inference, we experimented with retaining only top-K teacher and the corresponding student logits prior to computing $L_{logits}$. This should essentially remove noise from the low probability logits/tokens. We observe that a low value of top-K (<=100) results in a significant drop in accuracy. The drop is no longer observed when increasing top-K, but no better than not using top-K. Hence, we skip using top-K for further experiments.

## A.6 Choice of Losses

1. Using loss $L_o$ based on the **output activations** of encoder block provides a boost.
2. The final 1-2 layers in a Transformer for LLM are highly specialized [12] and mapping hidden states across **(last-2):(last-2)** layers for both the student and teacher achieves the best result [31].
3. Using **word embeddings** based loss($L_{emb}$) improves accuracy.
4. Computing loss $L_{att}$(attention relation loss [50]) based on query, key and value states does not show any improvement.
5. Adding loss $L_i$ based on the input to MLP makes no difference.
6. We weren't able to experiment with attention scores due to Flash Attention abstractions.
7. **Cosine similarity** loss performs the best.

Results are shown in Table 17.

## A.7 One-shot vs. Iterative Pruning and Distillation

| Loss components | LM loss |
|---|---|
| $L_{logits}$ | 2.155 |
| $L_{logits} + L_o(29:13)$ | **2.145** |
| $L_{logits} + L_o(15:15) + L_{emb}$ | 2.240 |
| $L_{logits} + L_o(23:15) + L_{emb}$ | 2.205 |
| $L_{logits} + L_o(29:15) + L_{emb}$ | 2.203 |
| $L_{logits} + L_o(30:15) + L_{emb}$ | 2.188 |
| $L_{logits} + L_o(31:15) + L_{emb}$ | 2.180 |
| $L_{logits} + L_o(28:12) + L_{emb}$ | 2.141 |
| $L_{logits} + L_o(\mathbf{29:13}) + L_{emb}$ | **2.141** |
| $L_{logits} + L_o(29:14) + L_{emb}$ | 2.152 |
| $L_{logits} + L_o(30:14) + L_{emb}$ | 2.150 |
| $L_{logits} + L_o(29:13) + L_{emb} + L_i(29:13)$ | 2.141 |

Table 17: Ablation study on loss components for computing $L_{is}$ and different (teacher:student) layer mapping for $L_o$ and $L_i$. LM loss is reported on the validation set of the 8T. Note: Layer indices start from 0, teacher Nemotron-4 15B layers (0-31), student MINITRON 8B-Depth-pruned layers (0-15).

**One-shot vs Iterative for Importance Estimation and Pruning**: refer to Table 12.

**One-shot vs Iterative within a Dimension:** to understand the best prune-retrain strategy considering a single dimension that can be pruned across the model (depth), we experiment with two different

| Loss | Tokens | MMLU | HellaSwag | HumanEval |
|------|--------|------|-----------|-----------|
| $L_{logits} + L_{is}$ | 18.9B | 58.0 | 73.6 | **26.8** |
| $L_{logits}$ | **18.9B** | **58.3** | **73.8** | 26.2 |
| $L_{logits}$ | 94B | 62.8 | 79.7 | 30.4 |

Table 18: Ablation study for MINITRON 8B with and without the loss component $L_{is}$, and increased retraining token count with $L_{logits}$. Adding $L_{is}$ performs on par with using $L_{logits}$ alone.

approaches for depth pruning and retraining in order to arrive at the MINITRON 8B-Depth-pruned model mentioned above.

As a first step, we rank layer importance with the procedure mentioned in 2.2 borrowed from [34]. Then we:

1. Iteratively prune and distill: Remove the least important layer, distill using 1.8B tokens and repeat the procedure 16 times. See $iterative \times 1\ 16l$ in Figure 7.
2. One-shot prune and distill: Remove 16 least important layers, distill using $1.8B \times 16(30.2B)$ tokens. See $1 - shot\ pruning\ 16l$ in Figure 7.

In order to mitigate the sharp drop in accuracy and to prevent further catastrophic collapse of the model, we increase the compute budget from 1.8B to 4 x 1.8B tokens starting with pruning of the 26 layer model which amounts to 86.4B tokens in total. See $iterative \times 4\ 16l$ in Figure 7. Increasing the training budget improves accuracy, but it still performs worse than the one-shot prune and distill strategy that uses 30.2B tokens.

With the iterative strategy, we can see in Figure 7 accuracy on:

- Hellaswag and PIQA is retained up to 31 layers and start dropping gradually with further removal of layers. We see a sharper drop when the model is reduced to 25 layers.
- MMLU score is retained up to 26 layers and start dropping gradually with further removal of layers. We see a sharp drop when the model is reduced to 20 layers.

**This shows that a few layers can be removed from a pretrained model in a lossless manner with minimal retraining. As for the retraining strategy, it is best to follow the one-shot method. Our results agree with [26].**

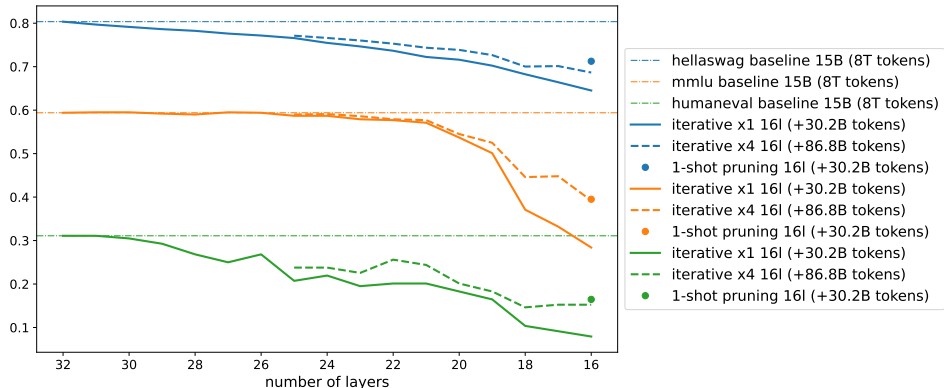

Figure 7: Accuracy on MMLU, HellaSwag and HumanEval benchmarks for iterative vs one-shot depth pruning and retraining strategy. One shot pruning and retraining outperforms the iterative approach.

**One-shot vs Iterative across Dimensions:** we experiment with iterative EMB→MLP-ATT pruning with retraining after both iterations and one-shot EMB-MLP-ATT pruning and retraining with equivalent token count as the former. As shown in Figure 8 one-shot achieves better results than the iterative approach.

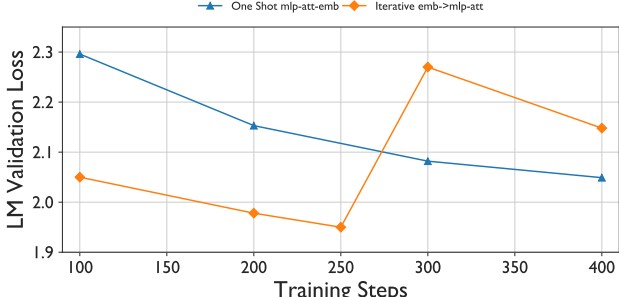

Figure 8: Comparison of LM validation loss curves for one-shot pruning of embeddings, MLP, attention and retraining for 400 steps vs the iterative approach; pruning the embeddings first and retraining for 250 steps, followed by pruning MLP, attention and retraining for additional 150 steps.

## A.8   Search

All the feasible 8B candidates produced by search are shown in Table 19.

| ID | Layers | Heads | MLP Exp. Factor | Embedding |
|----|--------|-------|-----------------|-----------|
| 1 | 32 | 32 | 12800 | 5120 |
| 2 | 32 | 32 | 13824 | 4608 |
| 3 | 32 | 48 | 11520 | 4608 |
| 4 | 32 | 48 | 16384 | 4096 |
| 5 | 31 | 32 | 12800 | 5120 |
| 6 | 31 | 32 | 16128 | 4608 |
| 7 | 31 | 48 | 13824 | 4608 |
| 8 | 31 | 48 | 16384 | 4096 |
| 9 | 30 | 32 | 12800 | 5120 |
| 10 | 30 | 32 | 16128 | 4608 |
| 11 | 30 | 48 | 13824 | 4608 |
| 12 | 30 | 48 | 16384 | 4096 |
| 13 | 29 | 32 | 12800 | 5120 |
| 14 | 29 | 32 | 16128 | 4608 |
| 15 | 29 | 48 | 13824 | 4608 |

Table 19: MINITRON 8B feasible candidates produced by search.

## A.9   Single vs. Multi-Phase Training

Table 20 compares the accuracy of single vs. multi-phase training.

| Strategy | Tokens | MMLU | HellaSwag | PIQA | HumanEval |
|----------|--------|------|-----------|------|-----------|
| Phase1 + Phase2 | 113B | 54.7 | 80.3 | 77.2 | 25.6 |
| **Phase2 only** | **94B** | **61.9** | 80.1 | 76.7 | **30.5** |

Table 20: Accuracy comparison of single vs multi-phase training approach with MINITRON 8B-Width-pruned. Note: This is not the searched 8B model in Table 2.

## A.10   Compute Resources

All experiments were performed on $16\times$ NVIDIA DGX A100 nodes ($8\times$ A100 80GB) for short turnaround times.

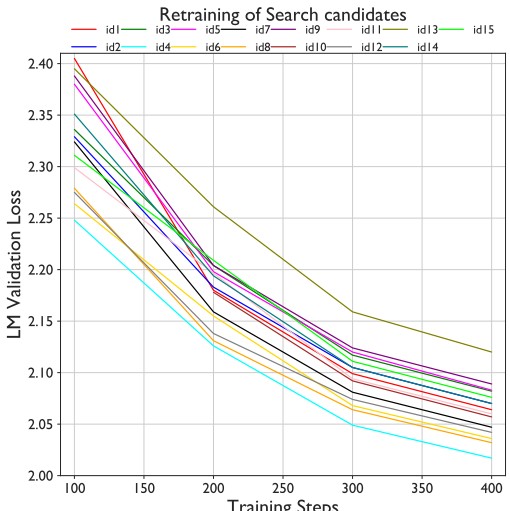

Figure 9: Retraining of searched candidates for 8B target with 1.8B training tokens.

