# OpenReview forum: "Compact Language Models via Pruning and Knowledge Distillation"
_NeurIPS.cc/2024/Conference — NeurIPS 2024 poster_

### Official Review · Reviewer_W3pm · 2024-07-07

**Soundness:** 3
**Presentation:** 3
**Contribution:** 2
**Rating:** 7
**Confidence:** 4

**Summary:**

This paper empirically explores compressing language models with pruning and knowledge distillation. It summarizes the best practices of pruning and distilling language models, which are supported by extensive experiments.

**Strengths:**

1. This paper is well-written, and the best practices are easy to follow, which is useful in practical LMs.
2. The paper includes sufficient experiments to support the main results (best practices).

**Weaknesses:**

1. The novelty of compressing LMs with pruning and knowledge distillation is limited because the method in this paper seems to be a simple combination of these two widely used techniques. Although the main contribution of this paper may be the best practices summarized from extensive experiments, it is better to highlight the difference between the final choice in this paper and the approaches in previous work like [1].
2. Extra computational cost should be considered. It seems the pruning process and the online inference of knowledge distillation requires extra computation. Besides the trained tokens, it is better to additionally compare the FLOPs of model training in the main experiments, like in Figure 1 and Table 2.


[1] Sheared LLaMA: Accelerating Language Model Pre-training via Structured Pruning. 2024. In ICLR.

**Questions:**

N/A

**Limitations:**

Suggestions:
The number of significant figures should remain consistent in Table 2 and Table 3.

---

> ### Author Rebuttal · Authors · 2024-08-06
>
> We would like to thank the reviewer for their encouraging comments and insightful feedback. Please find our responses below:
>
> >  **... it is better to highlight the difference between the final choice in this paper and the approaches in previous work like Sheared LLaMa.**
>
> To help with this comparison, we have created the following table to highlight the differences between our approach and Sheared LLaMa [1]:
>
> | Criteria    | Sheared LLaMa |  Minitron | Summary |
> | -------- | ------- | ------- | ------- |
> |  Pruning importance estimation | Learnable mask on  embedding, MLP and attention heads. Mask learning uses 0.4B tokens and is 5x slower than standard LM training [1]. | Forward pass on a tiny number of samples (1024 in paper) to compute importance for embedding, MLP and attention heads at once. This adds negligible overhead. | An expensive gradient-based importance estimation strategy (Sheared-LLaMa) is not required to achieve strong accuracy for pruned models. A simpler, forward pass only approach (Minitron) works well. |
> | Pruning dimensions | Depth, embedding, MLP, attention | Depth, embedding, MLP, attention | Both approaches support multiple width and depth axes. |
> | Retraining | Uses 50B tokens with conventional finetuning | Uses 94B tokens with knowledge distillation | Both papers find that retraining is required to recover accuracy loss on benchmarks. We showcase the superiority of knowledge distillation over conventional training and recommend the former as the retraining approach. |
> | Search | No search process. Matches a previously-defined fixed architecture configuration. | Ability to search for optimal architectures using a constrained brute-force algorithm. We also observe that lightweight retraining of searched candidates is essential to stabilize relative rankings. | Minitron can find better architectures through search. For eg: search finds that keeping attention heads intact is better for Minitron-8B. The paper also provides new insights on how zero-shot accuracy post-pruning isn’t reflective of final accuracy. |
> | Multiple compressed models | Requires repeating the 5x slower mask learning process N times for producing N compressed models. Each of the N models must also be finetuned. | Single importance estimation pass (negligible overhead) is sufficient for all N compressed models. Each of the N models must be distilled. | Minitron approach is significantly less costly when multiple compression targets are specified. |
>
> > **Extra computational cost should be considered. It seems the pruning process and the online inference of knowledge distillation requires extra computation. Besides the trained tokens, it is better to additionally compare the FLOPs of model training in the main experiments, like in Figure 1 and Table 2.**
>
> Importance estimation for pruning has negligible overhead, since we use a single forward pass on 1024 samples. This costs less than the forward pass on a single step for training the model, which uses 1152 samples.
>
> Regarding the overhead for knowledge distillation, we present ablations using a fixed amount of compute (trained on 128 GPUs for a fixed time) for conventional training vs distillation in Appendix Table 11 (with a 3B model, due to time constraints). We believe that this is more representative of real compute cost than FLOPs. Results show that distillation provides an absolute 1.1% gain in accuracy on the standard LM evaluation harness and a 5+% gain on MMLU when compared to conventional training.
>
> We also obtain the table below for a 4B model for consistency with the model sizes trained in the paper post submission. We plan to replace tables 11 and 12 in the paper with the one below. Rows marked with * below indicate settings with fixed compute/cost. In this table:
> 1. `4B-Random-Init` is a 4B model trained from scratch using conventional ground truth CE loss.
> 2. `4B-Pruned (prune Nemotron-4 15B)` is a 4B model pruned from a 15B model, using conventional ground truth CE loss for retraining.
> 3. `4B-Pruned-Distill (prune Nemotron-4 15B)` is a 4B model pruned from a 15B model, using distillation loss w.r.t. to the 15B teacher for retraining.
>
> | Model    | Tokens |  Hellaswag | MMLU |
> | -------- | ------- | ------- | ------- |
> | 4B-Random-Init  | 150B*    | 46.22 | 24.36 |
> | 4B-Random-Init |  400B     | 48.23 | 26.24 |
> | 4B-Pruned (Prune Nemotron-4 15B)   | 150B*  | 50.85 | 24.57 |
> | 4B-Pruned-Distill (Prune Nemotron-4 15B)   | 100B*   | 51.04 | 37.81 |
> | **4B-Pruned-Distill (Prune Minitron 8B)**   | **100B***    | **52.04** | **42.45** |
>
> We can see orthogonal improvements from 1 to 2, and 2 to 3 using a fixed amount of compute. This experiment demonstrates benefits of using knowledge distillation (+13% improvement in MMLU) with the additional computational cost for online inference factored in.
>
> > **The number of significant figures should remain consistent in Table 2 and Table 3.**
>
> Thank you for pointing this out, we will fix this in the final version.
>
> **References:**
> 1. Xia, Mengzhou, et al. "Sheared LLaMa: Accelerating language model pre-training via structured pruning." arXiv preprint arXiv:2310.06694 (2023).

---

> > ### Comment · Reviewer_W3pm · 2024-08-09
> > **Official Comment by Reviewer W3pm**
> >
> > I thank the authors for their response.
> >
> > After reading their reply, I will increase my score from 6 -> 7. I suggest the authors to include the additional experiment results in the final version of the paper.

---

> > > ### Author Response · Authors · 2024-08-09
> > >
> > > Thank you raising the score. Really appreciate it! We will make sure to include the additional results in the final paper.

---

### Official Review · Reviewer_enpQ · 2024-07-12

**Soundness:** 4
**Presentation:** 3
**Contribution:** 3
**Rating:** 7
**Confidence:** 4

**Summary:**

In this paper the authors explore compression of LLMs via pruning and Knowledge Distillation. They try out a variety of approaches for pruning as well as the retraining step and provide a comprehensive analysis of best practices for getting compact LLMs from their larger counterparts. The authors explore pruning across width and depth using importance scores and then retraining on a relatively small dataset via Knowledge distillation to get a capable compact model. This paper provides super interesting insights into model pruning which can be used in practice.

**Strengths:**

1. The paper is very well written and provide excellent explanations for all of their modeling choices.

2. I really enjoyed reading the key takeaways of the paper being structured as best practices. Each point in the best practices part of the paper provide some key insight into model compression and what works and what does not work.

3. I also enjoyed reading the comprehensive details that are written in the appendix.

Overall this is a very well thought out research paper with proper explanation for each modeling decision that is taken.

**Weaknesses:**

1. Small nitpick: The figures and tables need much much more detailed captions. I should have some idea about what the table wants to say just from the caption.

2. While results on benchmarks are appreciated, I always feel that these benchmarks do not reflect everything about LLMs. I would have liked some form of qualitative study on comparing the generations from the models. Some form of human evaluation on say 25-50 long form generation examples would be great given that there is not enough time during rebuttal for a larger scale human evaluation.

**Questions:**

1. In table 2 is the nemotron-8B baseline also trained on the 94B token retraining dataset? Even if the retraining dataset is a subset of the pretraining dataset of nemotron models, I believe the apples to apples comparison would be to retrain all the models on the retraining dataset as well. I understand retraining all the baselines would be tough given the time constraints but it would be great if you could retrain only the nemotron-8B model and share the results.

2. How does this compare to taking the retraining dataset, generate logits from nemotron-15B model and then using vanilla KD to train the nemotron-8B model further? So basically I ask for 2 more baselines if possible. First is further vanilla training on nemotron-8B on the retraining dataset and second is further KD on the nemotron-8B model with nemotron-15B as the teacher.

3. Check weakness 2

**Limitations:**

The authors have mentioned limitations.

---

> ### Author Rebuttal · Authors · 2024-08-07
>
> We would like to thank the reviewer for pointing out the strengths of the paper and providing valuable feedback.
>
> > **I would have liked some form of qualitative study on comparing the generations from the models. Some form of human evaluation on say 25-50 long form generation examples would be great ...**
>
> The reviewer raises an important question of evaluation via human preference. We conduct a small-scale (due to time constraints) human evaluation experiment with 7 people and 25 random prompts drawn from the MT-Bench multi-term benchmark. We generate responses from 3 models (all instruction-tuned): Minitron-4B, Gemma-2 2B, and Phi-2 2.7B and ask users to rate responses based on correctness, relevance, conciseness, hallucination and personal preference. For each prompt, the user selects a better model among Minitron and a randomly chosen second model (one of Gemma-2 or Phi-2), or specifies a tie. Model names are anonymized to prevent bias. The results of this experiment are shown in the table below:
>
> | Minitron  4B  | Tied | Other Model (Gemma-2 / Phi-2) |
> | -------- | ------- | ------- |
> | 43.43%  | 25.71%    | 30.86% |
>
> Here, we notice that Minitron is selected as a better model in 43.43% of cases. While this is a small-scale experiment at the moment, we will add a larger scale human evaluation to the final paper.
>
> Additionally, to gain more insights into the qualitative performance of Minitron models, we perform supervised fine-tuning (SFT) on Minitron 4B using instruction-tuning data used for Nemotron-4 to create Minitron 4B-instruct, and evaluate it on various tasks, including instruction-following and roleplay (IFEval and MT-Bench), RAG QA (ChatRAG-Bench), and function calling (BFCL). The results for this experiment are shown in Tables 6 to 9 in the 1-page PDF attached to the global rebuttal. Tables 6 to 8 demonstrate that Minitron 4B-instruct has strong instruction-following, roleplay and RAG capabilities, beating similarly sized models across all tasks. On function calling (Table 9), Minitron 4B-instruct outperforms Gemma-2B-IT and even Llama-3-8B-instruct. We will add these results to the final paper.
>
> > **In Table 2 is the nemotron-8B baseline also trained on the 94B token retraining dataset?**
>
> In Table 2, the nemotron-8B baseline is not trained on the exact 94B retraining dataset.
>
> > **So basically I ask for 2 more baselines if possible. First is further vanilla training on nemotron-8B on the retraining dataset and second is further KD on the nemotron-8B model with nemotron-15B as the teacher.**
>
> We agree that comparing models trained on exactly the same data is better. However, note that Nemotron-8B model is already trained on 3.8T tokens. Hence, we believe further vanilla training on the retraining dataset would be an unfair comparison and propose to train Nemotron-8B from scratch on the retraining dataset, identical to the retraining routine for Minitron-8B. This would provide an apples to apples comparison for: 1) training from scratch vs 2) our proposed approach, when training a new 8B model.
>
> Going the other way, we also have an ablation study where we train Minitron-8B on exactly the same Phase 1 dataset Nemotron-8B model was trained on:
>
> |    | Minitron 8B | Nemotron 8B |
> | -------- | ------- | ------- |
> |  Tokens |  **94.4B** | 3.5T |
> |  MMLU |   **0.521** | 0.485 |
> |  PIQA |   0.794 |  **0.7971** |
> |  HellaSwag |  **0.763** | 0.7587 |
> |  HumanEval |  **0.207** | 0.189 |
>
> Minitron outperforms Nemotron-8B on 3 out of 4 benchmarks above despite using a tiny random subset of the 3.8T dataset used by Nemotron 8B.
>
> As requested, we also perform the experiment with the model being trained with 3.8T + 94B tokens. For fairness, we also perform the exact same further vanilla training on Minitron 8B resulting in a model trained with 94B + 94B tokens.  The loss curves for the experiments are provided in Figure 1 in the 1-page PDF attached to the global rebuttal. Unfortunately, due to shortage of time, the new training jobs ran to 85% completion and we were unable to run downstream task evaluation.
>
> In the Figure:
> 1. Gray: Train nemotron-8B from scratch on the retraining dataset
> 2. Pink: Minitron 8B
> 3. Orange: Further vanilla training of nemotron-8B on the retraining dataset
> 4. Blue: Further vanilla training on minitron 8B on the retraining dataset
>
> 2 being significantly better than 1, and 4 being better than 3 showcases the efficacy of our technique.
>
> The experiment with KD on the Nemotron-8B model with Nemotron-15B as the teacher is not possible as these models use different tokenizers and therefore the logits will be misaligned between 2 models. Distillation of models with different tokenizers will be a great topic for future work.
>
> With the Nemotron-8B model (trained on 3.8T tokens) as a starting point, we would expect similar results to Minitron (Minitron pruning technique provides significantly better weight initialization compared to random initialization). Using the additional compute required to train Nemotron-8B on 3.8T tokens towards Minitron will make the latter even better.
>
> > **The figures and tables need much much more detailed captions**
>
> We absolutely agree that providing more details in the captions will help understand the underlying message. Some sample updated captions are provided below (unable to provide the full list of 12 changes due to space constraints, but will update them in the final paper):
>
> * Figure 2: High-level overview of our proposed iterative pruning and distillation approach to train a family of smaller LLMs. On a pretrained LLM, we first evaluate importance of neurons, rank them, trim the least important neurons and distill the knowledge from the original LLM to the pruned model. The original model is replaced with the distilled model for the next iteration of compression.
>
> * Figure 5: LM validation loss curve for retraining of two pruned candidates with (L2, L2) and (L2, Mean) metrics for (batch, sequence) aggregation strategies.

---

> > ### Comment · Reviewer_enpQ · 2024-08-07
> >
> > I appreciate the authors for adding the additional experiments and the qualitative study in such a short amount of time. According to me most weaknesses are addressed in the rebuttal. I am increasing my score to a 7.

---

> > > ### Author Response · Authors · 2024-08-08
> > >
> > > We'd like to thank the reviewer for raising their score. Truly appreciate it! We are very happy to see that our rebuttal has addressed your questions and concerns.

---

### Official Review · Reviewer_qFD8 · 2024-07-13

**Soundness:** 4
**Presentation:** 4
**Contribution:** 3
**Rating:** 7
**Confidence:** 4

**Summary:**

This paper proposes Compact Language Models via Pruning and Knowledge Distillation, which combines various tricks and methods to compress a 14B model to 8B while achieving better performance than training from scratch.

**Strengths:**

The paper conducts extensive experiments, comparing the latest baselines, and the authors summarize extensive tricks for pruning the model.

The pruned 8B model performs well, surpassing the model trained from scratch.

The paper is well-written, and the overall structure is good. Despite having many conclusions, it does not confuse the reader.

**Weaknesses:**

Will the authors open-source the code? If the code and data are open-sourced, I would raise my score.

When pruning a 70B model to 7B, would the method in the paper still work? Would this model perform better than pruning a 13B model to 7B?

**Questions:**

see weakness

---

> ### Author Rebuttal · Authors · 2024-08-06
>
> We would like to thank the reviewer for their encouraging comments and insightful feedback. Please find our responses below:
>
> > **Will the authors open-source the code? If the code and data are open-sourced, I would raise my score.**
>
> **Data:** more details on the composition of the Nemotron-4 dataset that we use were recently made public [1,2]. Parmar et al. [1] describe the distribution of domains, type of speech, toxicity, and quality of all our crawl data in Section 6, classifier information for all the above attributes in Appendix B, and information about the multilingual and programming language distribution on page 17. Parmar et al. [2] detail our continuous training recipe in Table 2, Figure 1, 2 and 4. Unfortunately, due to individual licenses for sub-data, we are unable to distribute the full dataset.
>
> **Code:**  the code is planned to be released as part of existing common libraries with a timeline of 1-2 months, with the first candidate being distillation with depth pruning, followed by width pruning (MLP, attention head, embedding). We are working on the modifications that will help apply the method to any existing models in Hugging face, and this requires refactoring.
>
> Models will also be released on HF with a permissive license. If required, we can also upload our checkpoints immediately to HF under an anonymous ID.
>
> > **When pruning a 70B model to 7B, would the method in the paper still work?**
>
> We observe that pruning should be performed iteratively in incremental steps. For example, as shown in Table 12 in the appendix, we observe an improvement of 4.6% points on MMLU by doing 15B->8B->4B instead of 15B->4B. Therefore one would expect better results by doing 70B->35B->15B etc.
>
> Pruning models of size 70B and bigger is definitely in our interest. This requires supporting _pipeline parallelization_ in addition to tensor parallelization. We are in the process of adding this functionality to the code-base and plan to release models as soon as they are available.
>
> **References:**
> 1. Parmar, Jupinder, et al. "Data, Data Everywhere: A Guide for Pretraining Dataset Construction." arXiv preprint arXiv:2407.06380 (2024).
> 2. Parmar, Jupinder, et al. "Reuse, Don't Retrain: A Recipe for Continued Pretraining of Language Models." arXiv preprint arXiv:2407.07263 (2024).

---

### Author Rebuttal · Authors · 2024-08-07

We sincerely thank all reviewers for their insightful feedback and comments. We have posted individual responses for each review, making every effort to provide additional results to support our responses within this limited rebuttal period. We hope our rebuttal addresses all reviewer concerns.

---

### Comment · Area_Chair_4nTD · 2024-08-12

Dear reviewers: as you are aware, the reviewer-author discussions phase ends on Aug 13. We request you to kindly make use of the remaining time to contribute productively to these discussions. If you have not read and/or responded to author rebuttal, please do it asap so that the authors get a chance to respond to you. If you have more questions to ask or want further clarification from the authors, please feel free to do it.

---

### Decision · Program_Chairs · 2024-09-25

**Decision:**

Accept (poster)

**Comment:**

The work explores combing model pruning and knowledge distillation for LLMs. The reviewers increased the scores after discussion and agreed to accept the submission. The engineering findings would be useful for practitioners. The first pruning stage seems critical in the whole pipeline. However, it would restrict applying the proposed method for cross-architecture distillation, i.e., the teacher model and student model use different model architectures. Overall, this is a good submission with a lot of details.